# Parasitoid biology preserved in mineralized fossils

Thomas van de Kamp [1], Achim H. Schwermann[2,3], Tomy dos Santos Rolo[4], Philipp D. Lösel[5],
Thomas Engler [3], Walter Etter[6], Tomáš Faragó[4], Jörg Göttlicher[4], Vincent Heuveline [5],
Andreas Kopmann [7], Bastian Mähler[3], Thomas Mörs[8], Janes Odar[4], Jes Rust[3], Nicholas Tan Jerome[7],
Matthias Vogelgesang[7], Tilo Baumbach[1,4] & Lars Krogmann[9,10]

About 50% of all animal species are considered parasites. The linkage of species diversity to a parasitic lifestyle is especially evident in the insect order Hymenoptera. However, fossil evidence for host–parasitoid interactions is extremely rare, rendering hypotheses on the evolution of parasitism assumptive. Here, using high-throughput synchrotron X-ray micro-tomography, we examine 1510 phosphatized fly pupae from the Paleogene of France and identify 55 parasitation events by four wasp species, providing morphological and ecological data. All species developed as solitary endoparasitoids inside their hosts and exhibit different morphological adaptations for exploiting the same hosts in one habitat. Our results allow systematic and ecological placement of four distinct endoparasitoids in the Paleogene and highlight the need to investigate ecological data preserved in the fossil record.

[1] Laboratory for Applications of Synchrotron Radiation (LAS), Karlsruhe Institute of Technology (KIT), Kaiserstr. 12, 76131 Karlsruhe, Germany. [2] LWL-Museum of Natural History, Sentruper Str. 285, 48141 Münster, Germany. [3] Steinmann Institute for Geology, Mineralogy and Palaeontology, University of Bonn, Nußallee 8, 53115 Bonn, Germany. [4] Institute for Photon Science and Synchrotron Radiation (IPS), Karlsruhe Institute of Technology (KIT), Hermann-von-Helmholtz-Platz 1, 76344 Eggenstein-Leopoldshafen, Germany. [5] Engineering Mathematics and Computing Lab (EMCL), Interdisciplinary Center for Scientific Computing (IWR), Heidelberg University, Im Neuenheimer Feld 205, 69120 Heidelberg, Germany. [6] Department of Geosciences, Natural History Museum Basel, Augustinergasse 2, 4051 Basel, Switzerland. [7] Institute for Data Processing and Electronics (IPE), Karlsruhe Institute of Technology (KIT), Hermann-von-Helmholtz-Platz 1, 76344 Eggenstein-Leopoldshafen, Germany. [8] Department of Palaeobiology, Swedish Museum of Natural History, Frescativägen 40, 114 18 Stockholm, Sweden. [9] Department of Entomology, State Museum of Natural History Stuttgart, Rosenstein 1, 70191 Stuttgart, Germany. [10] Institute of Zoology, Systematic Entomology, University of Hohenheim, 70593 Stuttgart, Germany. Correspondence and requests for materials should be addressed to T.v.d.K. (email: thomas.vandekamp@kit.edu) or to A.H.S. (email: achim.schwermann@lwl.org) or to L.K. (email: lars.krogmann@smns-bw.de)

Parasitic lifestyles are extremely successful among animals and evolved independently, perhaps hundreds of times[1]. With an estimated 50% of species, parasites comprise a huge proportion of animal life on Earth[2], and the arms races between parasites and their hosts are considered major driving forces for evolution[3]. In insects, parasitism is especially diverse in the order Hymenoptera, where many wasp species develop as parasitoids on or within an arthropod host, ultimately causing its death. In hymenopteran evolution, multiple transitions between host species, developmental stages and modes of parasitoidism are considered key events linked to enormous adaptive radiations[4–8] and an estimated 10–20% of all extant insects are parasitoid wasps[9,10]. Being antagonists of a wide variety of terrestrial arthropods, they have profound ecological and economic impact and many species are used as biological control agents[11,12].

Evidence for parasitism in fossils is generally rare[13], as it requires preserved information of interaction between both partners. As a consequence, the fossil record of parasitoid wasps is nearly exclusively restricted to isolated adults, with few examples of unidentified larvae trapped in amber next to their hosts[14–17]. Therefore, our understanding of parasitoid evolution is based on the inference that fossil organisms exhibited habits resembling those of their extant relatives. The only record of a putative fossil parasitoid wasp inside its preserved host derives from a thin-section of a mineralized fly pupa[18,19] from the later middle to late Eocene fissure fillings of the Quercy region in France, approximately 34–40 million years old[20]. The sectioned pupa was thought to comprise an adult braconid wasp, which was only traceable as faint silhouette lacking any diagnostic characters.

By employing robot-assisted synchrotron-based high-throughput X-ray microtomography, automated graphics processor unit (GPU)-based tomographic reconstruction and advanced semiautomated image segmentation algorithms, we investigate 1510 pupae of three different morphospecies sensu Handschin:[18] *Eophora* sp. (unavailable genus name[21]) (1448), *Megaselia* sp. (55) and *Spiniphora* sp. (15). The high number of specimens allow morphological and ecological characterization as well as systematic placement of endoparasitoid wasps. The parasitoids are identified as four new species of the family Diapriidae, which we assign to three genera, two of them new.

## Results

**Preservation and occurrence of parasitoids**. Externally, nearly all Quercy fly pupae were preserved as isolated endocasts, of which many were still covered by the puparium, the hardened skin of the last larval instar (Fig. 1a, j, Supplementary Fig. 1). Sometimes body parts of adult flies (especially legs) were recognizable through a partly translucent surface (Supplementary Fig. 1ay, be, Supplementary Table 1). Apart from legs and isolated bristles, remains of host flies (Fig. 1j–o) were rarly preserved and did not provide diagnostic characters. In 55 pupae (3.8%) of *Eophora* we identified parasitation events, which were mostly represented by adult wasps. Preservation of the parasitoids ranged from barely recognizable to well-preserved specimens (Supplementary Figs. 2–5, Supplementary Table 1). In most cases, sclerites were preserved as voids inside the mineralized matrix (Fig. 1c, g–i). Nineteen wasps had folded wings and showed the symmetric posture of a late wasp pupa (Fig. 1e), while 20 specimens were evidently hatched, as indicated by unfolded wings and an asymmetric body posture (Fig. 1f, Supplementary Table 1).

**Systematic palaeontology**. In order to assess the size variation within the species, we measured the length between the anterior margin of the propleurae and the anterior tip of the median keel of the propodeum. The reference lengths for holo- and paratypes are listed in Supplementary Table 1 along with information on the preservation of hosts and parasitoid wasps. All other measurements refer to holotypes only and are included in the species descriptions. Differences in size and preservation are further documented by surface renderings of 30 parasitoid heads covering all species (Supplementary Fig. 5).

Family Diapriidae Haliday, 1833
Subfamily Diapriinae Haliday, 1833
Tribe Spilomicrini Ashmead, 1893
*Xenomorphia* Krogmann, van de Kamp & Schwermann
gen. nov.

**Type species**. *Xenomorphia resurrecta* Krogmann, van de Kamp & Schwermann sp. nov.
**Etymology**. The genus name refers to the endoparasitoid Xenomorph creature featured in the "Alien" media franchise.
**Diagnosis**. Antenna 14-segmented in both sexes, apical flagellomeres bead-like. Epistomal sulcus distinct and straight. Mandibles narrow, 2-toothed. Labrum exposed.

*Xenomorphia resurrecta* Krogmann, van de Kamp & Schwermann sp. nov.

**Etymology**. The specific epithet points out the "resurrection" of the extinct species by means of digital imaging.
**Diagnosis**. Malar sulcus distinct. Petiole cylindrical, 1.6–1.8 times as long as wide.
**Referred material**. Holotype ♀: NMB F2875. Paratypes ♀♀: NMB F2615, NMB F2822, NMB F2840, NMB F2856, NMB F2972, NMB F2982, NMB F3018, NMB F3103, NMB F3220, NMB F3389, NMB F3394, NMB F3477, NMB F3612, NRM-PZ Ar65771, NRM-PZ Ar65793, NRM-PZ Ar65913, and NRM-PZ Ar65938. Paratypes ♂♂: NMB F2557, NMB F2674, NMB F2732, NMB F2752, NMB F2831, NMB F2851, NMB F2854, NMB F2945, NMB F2985, NMB F3140, NMB F3146, NMB F3254, NMB F3278, NMB F3516, NMB F3562, NMB F3610, NRM-PZ Ar65720, NRM-PZ Ar65767, NRM-PZ Ar65772, NRM-PZ Ar65794, NRM-PZ Ar65800, NRM-PZ Ar65823, NRM-PZ Ar65895, and NRM-PZ Ar65948 (Figs. 1d–f, 2, 3, 4, Supplementary Fig. 2, Supplementary Fig. 3, Supplementary Fig. 5a–x, Supplementary Data 1 and 2, Supplementary Movies 1 and 2).
**Locality**. The fossils originate from the phosphorite mines of the Paleogene fissure fillings of the Quercy region in South-Central France. The specimens were discovered near Bach[18] (coordinates: 44°21′ N, 1°40′ E). More information on the exact locality, collection date and collector are unknown.
**Description**. Female (Figs. 1f, 3, 4a–m). Reference length: 824 μm. Head sculpture mainly smooth, frons with scattered setiferous punctures. Head height: 529 μm, head width: 533 μm, and head length: 435 μm. Ocelli large, ratio between interocellar distance and oceloocular distance (IOD:OOD) = 0.58. Eyes large, 277 μm high and 215 μm wide. Malar space 91 μm, malar sulcus present. Occipital carina complete, horseshoe-shaped, anteriorly marked by small ridges. Clypeus narrow, laterally with enlarged anterior tentorial pits, dorsally marked by distinct and straight epistomal sulcus. Mandibles narrow, two-toothed, leaving large, semicircular area from ventral clypeal margin. Area covered by membranous labrum. Toruli oriented dorsally, positioned on distinct antennal shelf, about half-way upon face. Antennal shelf with transverse wrinkles. Antennal shelf connected to epistomal sulcus by two submedian frontal sulci. Supraclypeal area between sulci slightly expanded. Antenna: 14-segmented, elbowed with

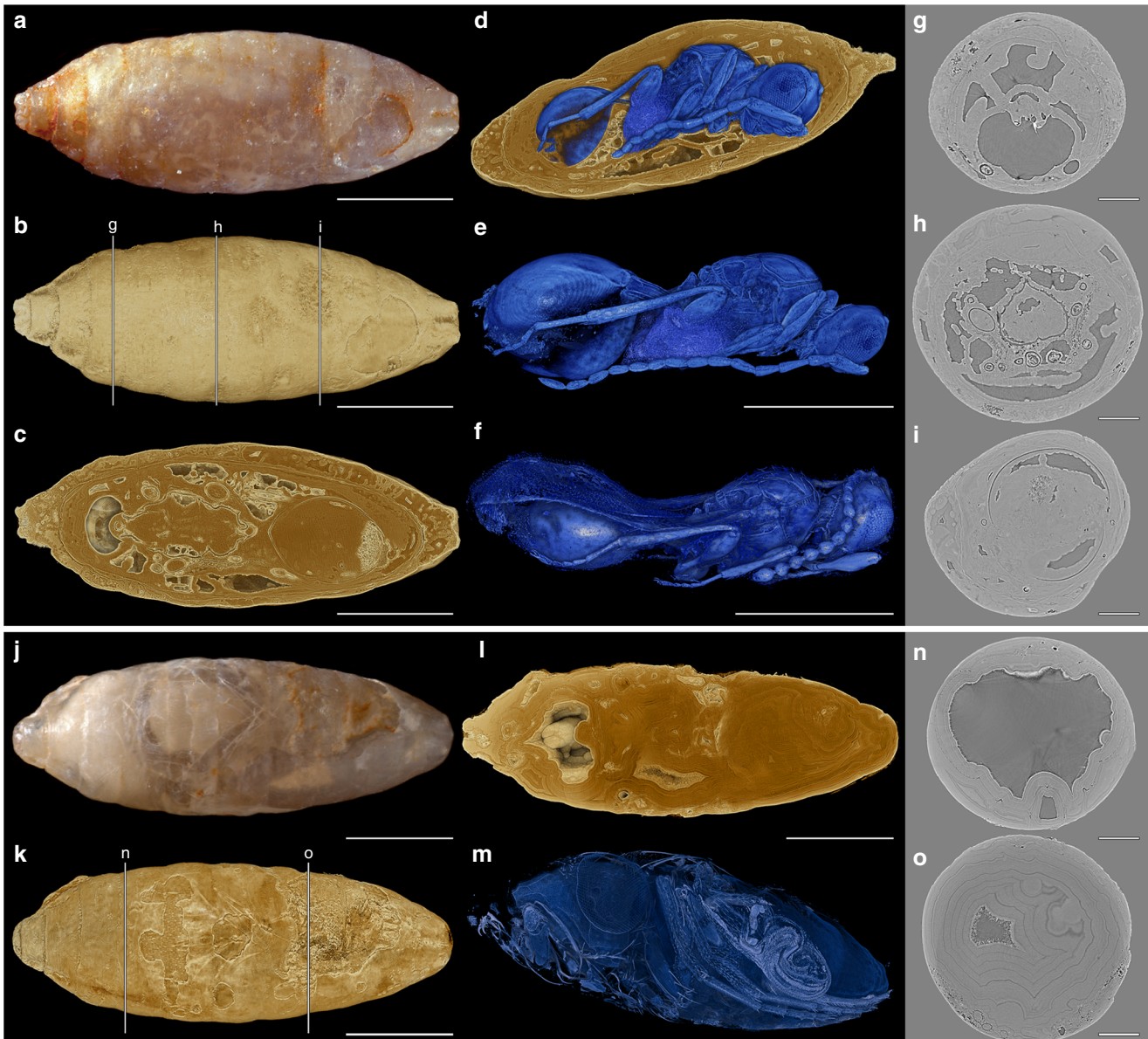

**Fig. 1** X-ray tomography of fossils. **a** Host fly puparium (NRM-PZ Ar65720). **b** Volume rendering of **a**. **c** Longitudinal section of **b**. **d** Parasitoid wasp inside host (perspective view; Supplementary Movie 1). **e** *Xenomorphia resurrecta*, male (NRM-PZ Ar65720) in symmetric posture with folded wings. **f** *X. resurrecta*, female (NMB F2875) in asymmetric posture with unfolded wings. **g–i** Transverse sections of tomogram as indicated in **b**. **j** Host fly puparium (NRM-PZ Ar65810). **k** Volume rendering of **j**. **l** Longitudinal section of **k**. **m** Volume rendering of host fly (perspective view). **n–o** Transverse sections of tomogram as indicated in **k**. Scale bars: **a–c**, **e**, **f**, **j–l** = 1 mm; **g–i**, **n**, **o** = 250 µm

elongate scape. Scape reaching mid-height of lateral ocellus. Apical flagellomeres bead-like, with few scattered setae. First flagellomere cylindrical, distinctly longer than subsequent flagellomeres. Subsequent flagellomeres short, hardly longer than wide. Multiple setal bases present as pores on individual antennomeres. Pronotum with distinct neck, dorsal pronotal surface short, and pronotum adjacent to mesoscutum. Posterior pronotal margin with elongate setae. Lateral panel of pronotum large and triangular, adjacent to mesopleuron. Pronotal depression for accommodation of profemur absent. Pronotal neck with irregular sculpture, dorsal and lateral pronotal margin with indistinct foveae, rest of pronotum smooth. Hind corner of pronotum reaching tegula. Mesothoracic spiracles positioned at lateral margin of pronotum, posteriorly enclosed by prepectal shelf, dorsally reaching tegula. Mesothoracic spiracles nearly completely enclosed by cuticle, omitting just small membranous stripe dorsally. Prosternum subrectangular, transversely divided by complete cross carina. Three large profurcal pits present, one median in anterior half and two submedian in posterior half of prosternum. Profurca u-shaped, profurcal arms completely fused with prosternum. Position of articulation point between pro-pleuron and profurcal arm at posterior end of propectus. Propleural arms anteriorly pointed.

Mesoscutum smooth, with few scattered elongate setae. Mesoscutal suprahumeral sulcus weakly developed. Notauli present as broad, curved sulci, which are slightly dilated posteriorly. Notauli anteriorly nearly reaching anterior mesoscutal margin and posteriorly nearly reaching transscutal articulation. Notauli internally preserved as rather sharp ridges. Transscutal articulation straight and complete. scutoscutellar

sulcus marked by two large, ovoid pits, which are medially separated by straight ridge. Pits internally not well marked. Axillae narrow and smooth. Axillulae with two rows of short setae. Mesoscutellar disc laterally separated from axillula by short ridges. Hind margin of mesoscutellum distinctly foveolate. Mesopleuron smooth and glabrous, mesofemoral depression indistinct. Mesopleuron laterally divided by diagonal sulcus. Mesepisternum anteriorly with distinct procoxal depressions, which are medially separated by distinct carina. Acetabular and mesotrochantinal carina present, meeting medially on ventral mesopleuron. Mesodiscrimen complete and foveolate. Anterior mesofurcal pit inconspicuous, marked by anteriormost fovea of foveolate mesodiscrimen. Posterior mesofurcal pit present between mesocoxal foramina. Mesocoxal foramina not completely enclosed by cuticle. Mesodiscrimenal lamella not reaching anterior margin of mesopectus. Anterior mesofurcal base situated about mid-way through median mesopectal length. Mesofurcal bridge medially interrupted, situated only slightly above mid-height of mesofurca.

Metascutellum with two raised lateral and one raised median carina, and one less distinct transverse carina, metascutellum posteriorly expanded. Lateral panel of metanotum composed of anteriorly reduced foveae. Metapleuron subrectangular, coarsely reticulate. Metepisternum with distinct depressions for accommodating mesocoxae but without transverse or median carina. Single metafurcal pit present anteromedially of metacoxal foramina. Metafurca indistinct, u-shaped, basally fused to highly raised paracoxal ridge. Metadiscrimenal lamella reaching mid-level of metacoxal foramina.

Propodeum with coarse irregular sculpture, medially with anteriorly projecting keel. Plicae and median carina present. Hind margin of propodeum carinate. Petiole cylindrical, laterally with short pilosity, 1.64 times as long as wide. Petiole dorsally with multiple irregular longitudinal carinae, ventrally with fewer longitudinal carinae. Second metasomal tergum enlarged, anterior margin medially divided, overlapping petiole. Subsequent terga short. Second and third metasomal sternum enlarged, subsequent sterna short.

Wings: Forewing unfolded, venation not traceable, and outer wing margin with long pilosity.

Legs: Foreleg with elongate simple trochanter, protibial spur with distinct cleft. Midleg with elongate simple trochanter and two mesotibial spurs. Hind leg with two-segmented trochanter and two metatibial spurs.

Male (Figs. 1e, 4n–v). Measurements given for paratype: NRM-PZ Ar65720. Very similar to female but differs in following features. Reference length: 911 μm. Head height: 605 μm, head width: 599 μm, and head length: 481 μm. IOD:OOD: 0.70. Antenna: 14-segmented, but distinctly longer than in female. Scape, pedicel, and first flagellomere comparable to female, but subsequent flagellomeres cylindrical, i.e., distinctly longer than broad. Eyes 328 μm high and 260 μm wide. Malar space 82 μm. Petiole very similar in proportions (1.75 times as long as wide) and shape, but with more extensive pilosity, also extending to ventral surface. All legs with two-segmented trochanters.

### Xenomorphia handschini Krogmann, van de Kamp & Schwermann sp. nov.

**Etymology**. The species epithet honors Swiss entomologist Eduard Handschin (1894–1962), who found the first traces of a parasitoid wasp in the Quercy fossils and recognized the scientific importance of these deposits.

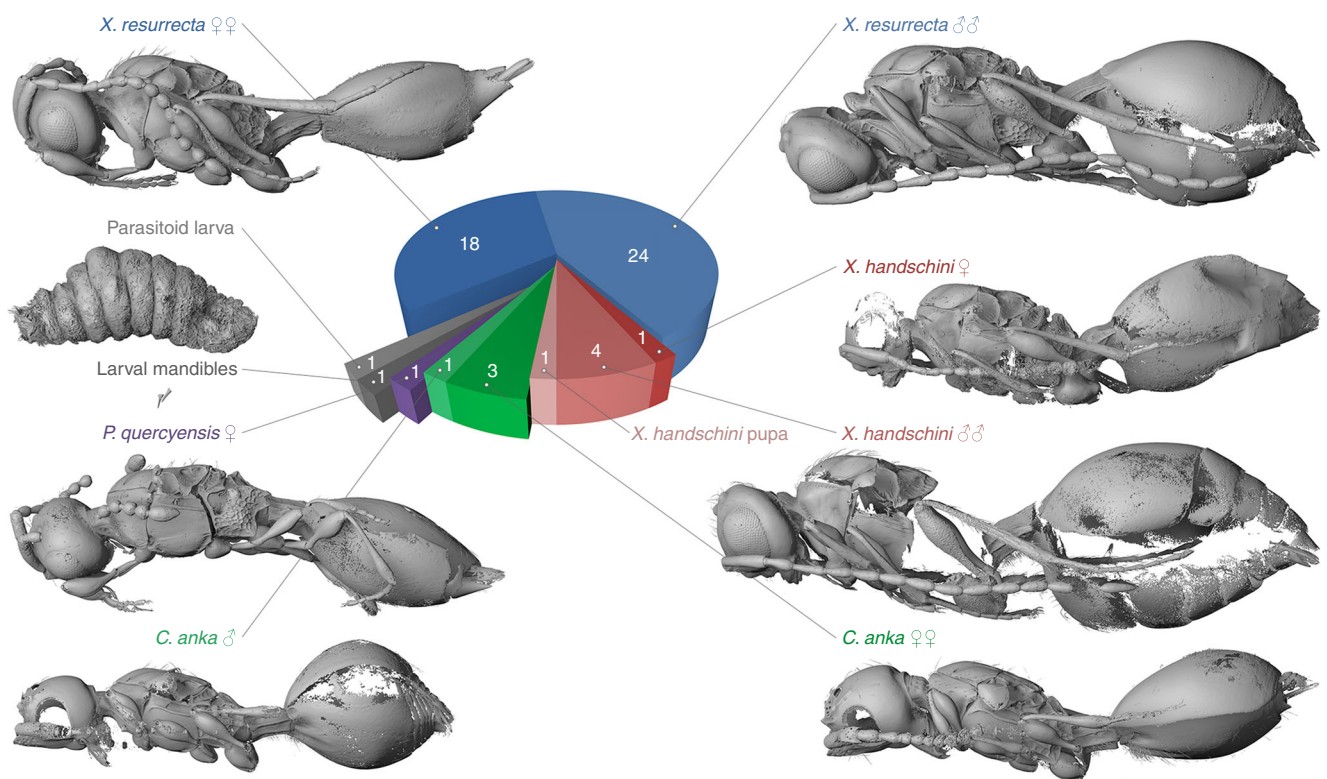

**Fig. 2** Visualization and frequency of the four parasitoid species. A total number of 55 parasitation events were recognized. *Xenomorphia resurrecta* dominated with 18 females and 24 males. *Xenomorphia handschini* was represented by one female, three males, and one pupa of undetermined sex (not displayed), *Coptera anka* by three females and one male and *Palaeortona quercyensis* by a single female. Additionally, a single unidentified putative second instar larva and a set of last larval instar mandibles presumably left behind by an emerged parasitoid were identified. Scale bar: 1 mm

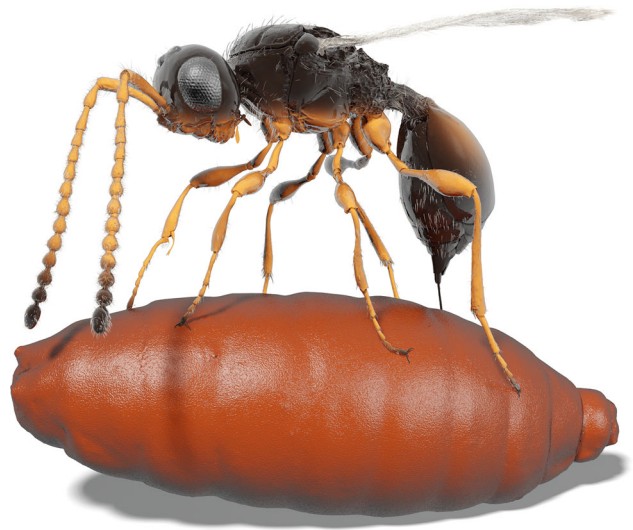

**Fig. 3** Illustration of a female *Xenomorphia resurrecta* ovipositing into a puparium. The picture is directly based on tomography data of NMB F2875 (wasp; Figs. 1f, 2, 4a–m, Supplementary Figs. 1e, 2e, and 5b) and NRM-PZ Ar65767 (puparium; Supplementary Fig. 1aj). Colors and parts of pilosity are imaginary. Supplementary Movie 2 shows how the illustration was derived from original tomography data

**Diagnosis.** Malar sulcus distinct only in males, faintly indicated in females. Petiole distinctly broadened, subquadrate, 1.03–1.20 times as long as wide.

**Referred material.** Holotype ♀: NMB F3042. Paratypes ♂♂: NMB F2543, NMB F3192, NMB F3571, and NRM-PZ Ar65942. Pupa: NRM-PZ Ar65822 (Figs. 2, 5, Supplementary Fig. 4a–f, Supplementary Fig. 5y–aa, Supplementary Data 3, 4).

**Locality.** As for *X. resurrecta*.

**Description.** Female (Fig. 5a–m). Reference length: 734 μm. Head sculpture mainly smooth, frons with scattered punctures. Setae not preserved. Head height: 467 μm, head width: 483 μm, head length: 396 μm. Ocelli large. Eyes large, 242 μm high and 209 μm wide. Malar space 66 μm, malar sulcus absent. Occipital carina complete, horseshoe-shaped, anteriorly marked by small ridges. Clypeus narrow, laterally with enlarged anterior tentorial pits, dorsally marked by distinct and straight epistomal sulcus. Mandibles covered by numerous scattered punctures. Mandibles narrow, two-toothed, leaving large, semicircular area from ventral clypeal margin. Area covered by membranous labrum. Toruli dorsally oriented, positioned on indistinct antennal shelf, about half-way upon face. Antennal shelf with few indistinct oblique wrinkles. Antennal shelf connected to epistomal sulcus by two submedian frontal sulci. Supraclypeal area between sulci slightly expanded. Antenna: 14-segmented, elbowed with elongate scape. Scape reaching mid-height of lateral ocellus. Apical flagellomeres bead-like, with few scattered setae. First flagellomere longer than subsequent flagellomeres. Subsequent flagellomeres distinctly to slightly longer than wide. Multiple setal bases present as pores on individual antennomeres. Pronotum with distinct neck, dorsal pronotal surface short, pronotum adjacent to mesoscutum. Posterior pronotal margin with elongate setae. Lateral panel of pronotum large and triangular, adjacent to mesopleuron. Pronotal depression for accommodation of profemur absent. Pronotal neck with irregular sculpture, dorsal and lateral pronotal margin with indistinct foveae, rest of pronotum smooth. Hind corner of pronotum reaching tegula. Mesothoracic spiracles positioned at lateral margin of pronotum, posteriorly enclosed by prepectal shelf, dorsally reaching tegula. Mesothoracic spiracles nearly completely enclosed by cuticle omitting just small membranous stripe dorsally. Prosternum subrectangular, transversely divided by complete cross carina. Three large profurcal pits present, one median in anterior half and two submedian in posterior half of prosternum. Profurca u-shaped. Only bases of profurcal arms preserved. Propleural arms incompletely preserved.

Mesoscutum smooth, with numerous scattered elongate setae. Notauli present as very broad, curved sulci, which are slightly dilated posteriorly. Notauli anteriorly nearly reaching anterior mesoscutal margin and posteriorly nearly reaching transscutal articulation. Notauli internally marked by broad ridges. Transscutal articulation straight and complete. Scutoscutellar sulcus marked by two large semicircular pits, which are medially separated by straight ridge. Pits internally well marked. Axillae narrow and smooth. Axillulae with scattered short setae. Mesoscutellar disc laterally separated from axillula by short ridges. Hind margin of mesoscutellum distinctly foveolate. Mesopleuron smooth and glabrous, mesofemoral depression indistinct. Mesopleuron laterally divided by diagonal sulcus. Mesepisternum anteriorly with distinct procoxal depressions, which are medially separated by distinct carina. Acetabular carina only weakly indicated, mesotrochantinal carina distinct, both carinae meeting medially on ventral mesopleuron. Mesodiscrimen complete and foveolate. Single mesofurcal pit present between mesocoxal foramina. Mesocoxal foramina not completely enclosed by cuticle. Mesofurca not preserved.

Metascutellum with two raised lateral and one raised median carina, and one less distinct transverse carina, metascutellum posteriorly expanded. Lateral panel of metanotum mainly smooth with traces of reduced foveae. Metapleuron subrectangular, coarsely reticulate. Metepisternum with indistinct depressions for accommodating mesocoxae, and without transverse or median carina. Single metafurcal pit present anteromedially of metacoxal foramina. Metafurca not preserved.

Propodeum with coarse irregular sculpture, medially with anteriorly projecting keel. Plicae and median carina present. Hind margin of propodeum carinate. Petiole distinctly broadened, subquadrate, 1.03 times as long as wide, laterally with short pilosity. Petiole dorsally with multiple irregular longitudinal carinae, ventrally mainly smooth, longitudinal carinae only posteriorly indicated. Second metasomal tergum enlarged, anterior margin medially divided, overlapping petiole. Subsequent terga short. Second and third metasomal sternum enlarged, subsequent sterna short.

Wings: Folded but hardly traceable.

Legs: All legs with two-segmented trochanters. Protibial spur with distinct cleft. Midleg with two mesotibial spurs. Hind leg with two metatibial spurs.

Male (Fig. 5n–v). Measurements given for paratype: NMB F2543. Very similar to female but differs in following features. Reference length: 911 μm. Head height: 596 μm, head width: 608 μm, head length: 478 μm. IOD:OOD: 0.74. Frons with scattered punctures and elongate setae. Antenna: 14-segmented, but distinctly longer than in female. Scape, pedicel and first flagellomere comparable to female, but subsequent flagellomeres cylindrical, i.e., distinctly longer than broad. Multiple setal bases present as pores on individual antennomeres and distinct setation preserved. Eyes 321 μm high and 243 μm wide. Malar space 93 μm, malar sulcus present. Mandibles smooth, without scattered punctures. Antennal shelf with few oblique wrinkles. Scutoscutellar sulcus marked by two large irregular pits, which are medially separated by two oblique ridges. Axillulae with distinct short setation. Acetabular carina distinctly indicated. Petiole 1.20 times as long as wide, laterally with scattered elongate seta and ventrally with dense short setation. Few complete longitudinal carinae present ventrally.

Comments: Both species of *Xenomorphia* share 14-segmented antennae in both sexes with the extant genera *Xenismarus* Ogloblin and *Chilomicrus* Masner and García. These genera are restricted to South America and considered early lineages of Diapriidae[22]. *Xenomorphia* can be readily distinguished from *Chilomicrus* in a number of morphological characters, including presence of an exposed labrum (absent in *Chilomicrus*), apical flagellomeres bead-like (subquadrate in *Chilomicrus*) and

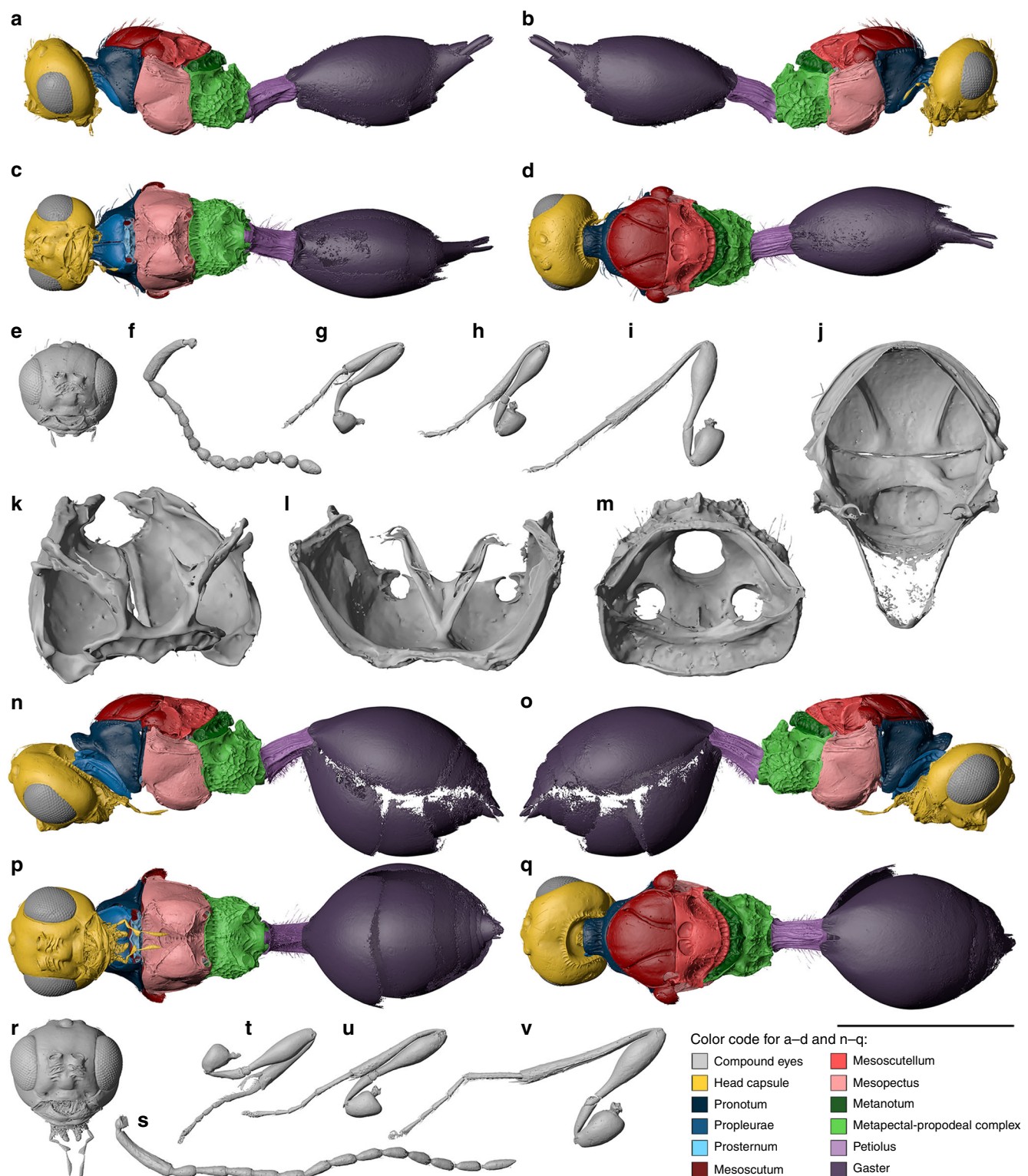

**Fig. 4** Digital reconstruction of *Xenomorphia resurrecta* gen. & sp. nov. (Diapriidae: Diapriinae: Spilomicrini). Female holotype NMB F2875 (**a–m**) and male paratype NRM-PZ Ar65720 (**n–v**). Habitus (**a–d**), head (**e**), left antenna (**f**), left foreleg (**g**), right midleg (**h**), right hind leg (**i**). Internal anatomical structures (**j–m**): mesonotum ventral view (**j**), propectus dorsolateral view (**k**), mesopectus anterior view (**l**), metapectal-propodeal complex anterior view (**m**). Habitus (**n–q**), head (**r**), left antenna (**s**), right foreleg (**t**), right midleg (**u**), right hind leg (**v**). Scale bar: **a–i**, **n–v** = 1 mm; **j** = 500 μm; **k** = 250 μm; **l**, **m** = 400 μm

Color code for a–d and n–q:

- Compound eyes
- Head capsule
- Pronotum
- Propleurae
- Prosternum
- Mesoscutum
- Mesoscutellum
- Mesopectus
- Metanotum
- Metapectal-propodeal complex
- Petiolus
- Gaster

scutoscutellar sulcus marked by two large, ovoid pits, which are medially separated by a straight ridge (not clearly separated in *Chilomicrus*). *Xenomorphia* appears morphologically very similar to *Xenismarus* but differs in having the upper tooth of the mandible shorter than the lower tooth and in the presence of plicae. As most shared characters seemingly represent symplesiomorphies, we here refrain from placing the fossil species in the extant genus *Xenismarus* based on the available morphological evidence, but it may turn out to be closely related or even congeneric.

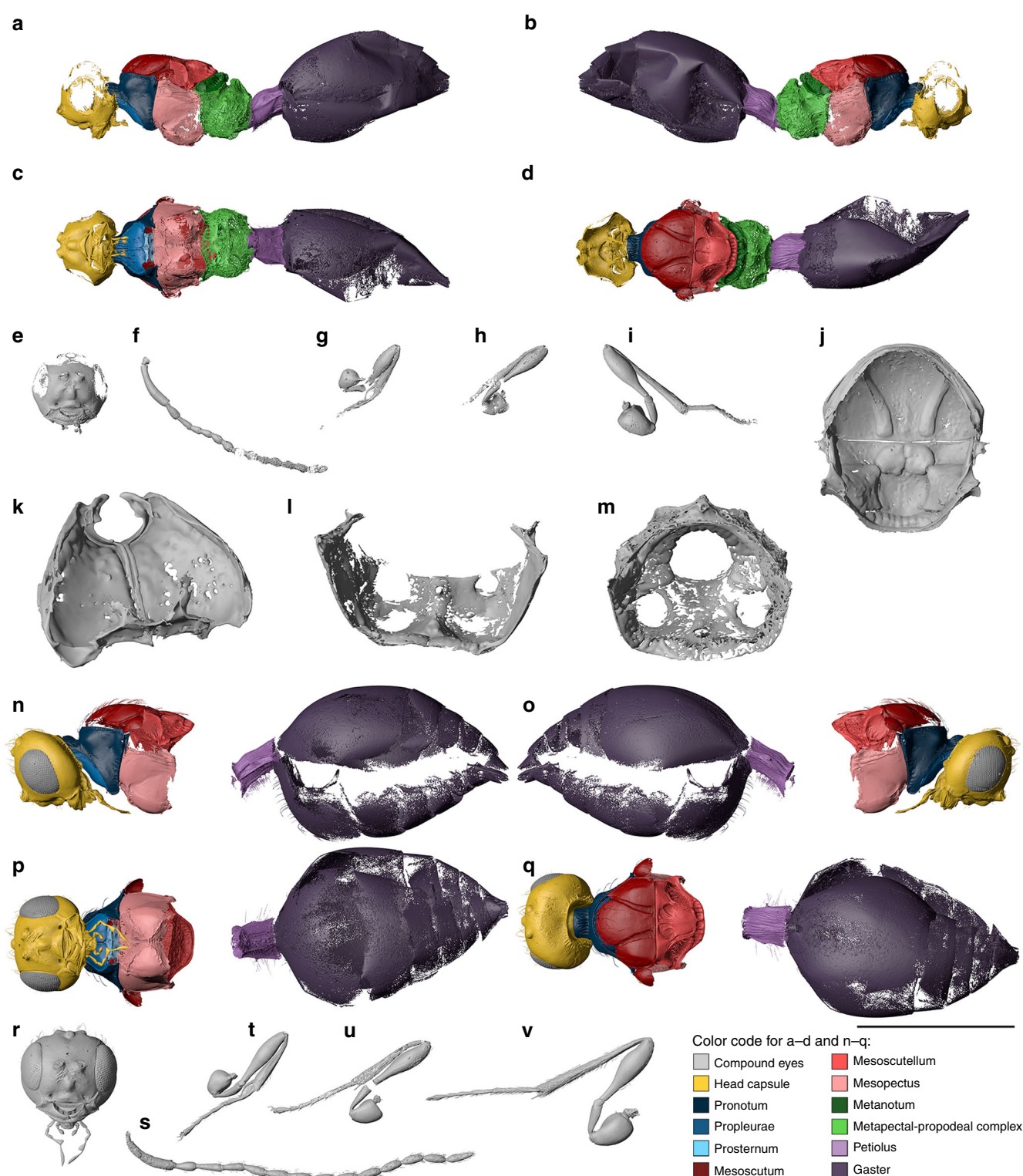

**Fig. 5** Digital reconstruction of *Xenomorphia handschini* gen. & sp. nov. (Diapriidae: Diapriinae: Spilomicrini). Female holotype NMB F3042 (**a–m**) and male paratype NMB F2543 (**n–v**). Habitus (**a–d**), head (**e**), left antenna (**f**), right foreleg (**g**), right midleg (**h**), left hind leg (**i**). Internal anatomical structures (**j–m**): mesonotum ventral view (**j**), propectus dorsolateral view (**k**), mesopectus anterior view (**l**), metapectal-propodeal complex anterior view (**m**). Habitus (**n–q**), head (**r**), right antenna (**s**), right foreleg (**t**), right midleg (**u**), right hind leg (**v**). Scale bar: **a–i**, **n–v** = 1 mm; **j** = 500 μm; **k** = 250 μm; **l**, **m** = 400 μm

Tribe Psilini Hellén, 1963
*Coptera* Say, 1836

Postgenal cushion present. For full diagnosis see Masner and García[22].

**Diagnosis**. Body predomimantly smooth. Labrum exposed, subtriangular. Mandible long, falcate, projecting posteriorly.

*Coptera anka* Krogmann, van de Kamp & Schwermann sp. nov.

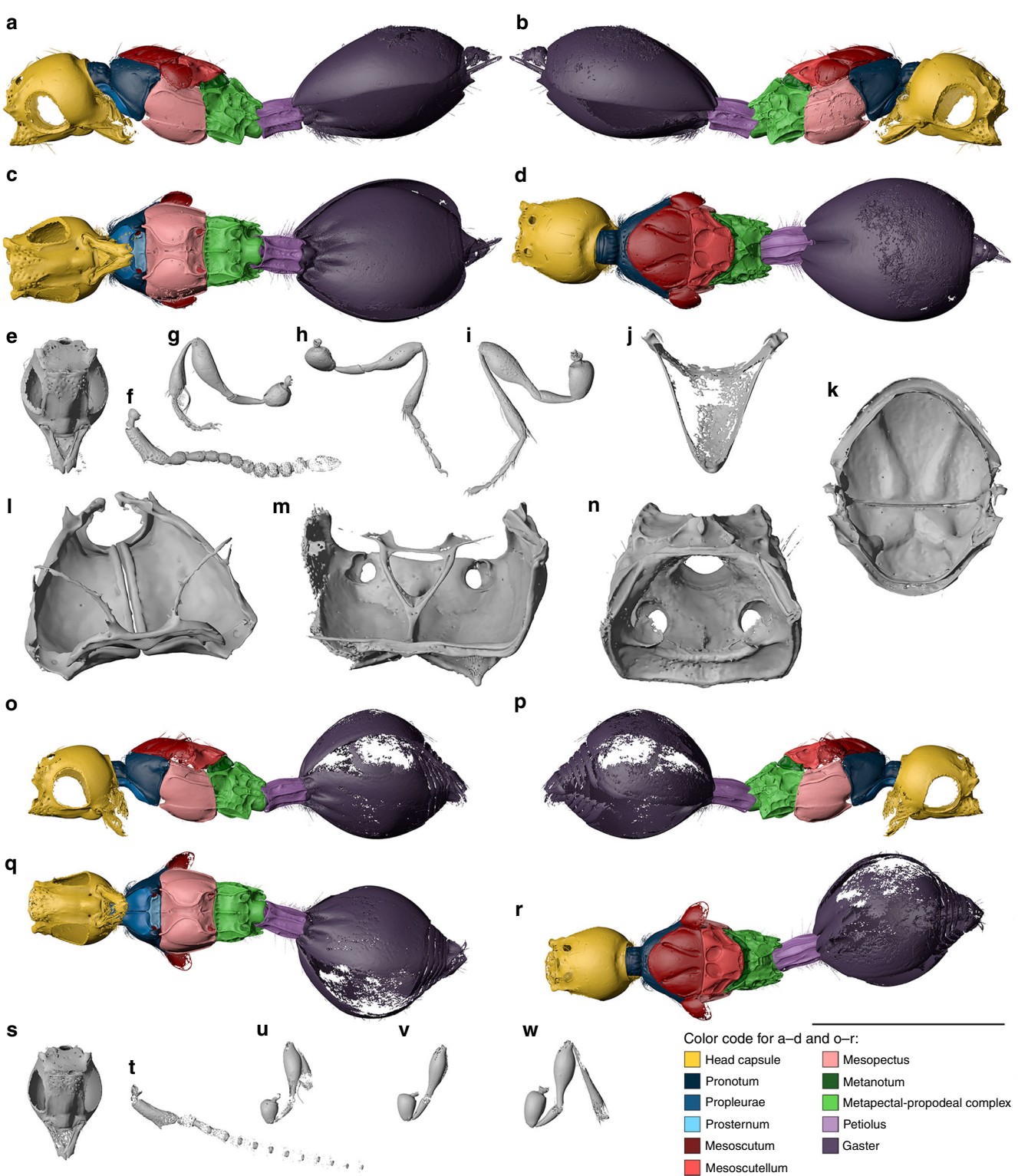

**Fig. 6** Digital reconstruction of *Coptera anka* sp. nov. (Diapriidae: Diapriinae: Psilini). Female holotype NRM-PZ Ar65897 (**a–f**, **k**, **l**), female paratype NMB F2848 (**g–j**, **m**, **n**) and male paratype NMB F3154 (**o–w**). Habitus (**a–d**), head (**e**), left antenna (**f**), left foreleg (**g**), left midleg (**h**), right hind leg (**i**). Internal anatomical structures (**j–m**): mesophragma ventral view (**j**), mesonotum ventral view (**k**), propectus dorsolateral view (**l**), mesopectus anterior view (**m**), metapectal-propodeal complex anterior view (**n**). Habitus (**o–r**), head (**s**), right antenna (**t**), right foreleg (**u**), left midleg (**v**), left hind leg (**w**). Scale bar: **a–i**, **o–w** = 1 mm; **j**, **k** = 500 µm; **l** = 250 µm; **m**, **n** = 400 µm

Color code for a–d and o–r:
- Head capsule
- Pronotum
- Propleurae
- Prosternum
- Mesoscutum
- Mesoscutellum
- Mesopectus
- Metanotum
- Metapectal-propodeal complex
- Petiolus
- Gaster

**Etymology**. The specific epithet is the Swedish word for duck and refers to the duck-like appearance of the head and the origin of the holotype from the Swedish Museum of Natural History.

**Diagnosis**. Head distinctly elongate anteriorly with toruli at anteriormost position. Ocelli large, ratio between IOD and OOD 1.11. Mandible three-toothed.

**Referred material**. Holotype ♀: NRM-PZ Ar65897. Paratypes ♀♀: NMB F2848, NMB F2954. Paratype ♂: NMB F3154 (Figs. 2 and 6, Supplementary Fig. 4g–j, Supplementary Fig. 5ab, ac, Supplementary Data 5 and 6).

**Locality**. As for *X. resurrecta*.

**Description**. Female (Fig. 6a–n). Reference length: 717 µm. Body predominantly smooth. Head distinctly elongate anteriorly. Head with deep punctures on antennal shelf and on frons anterior to ocelli. Head height: 523 µm, head width: 456 µm, head length: 592 µm. Ocelli large, IOD:OOD = 1.11. Eyes large, 261 µm high and 210 µm wide. Malar space 67 µm, malar sulcus absent. Occipital carina complete, semicircular, anteriorly not marked by small ridges or punctures. Clypeus convex, about as high as wide, laterally with distinct anterior tentorial pits, dorsally hardly marked by faint epistomal sulcus. Mandibles extremely narrow and elongate, not clasped, 3-toothed, projecting diagonally backward. Labrum triangular. Oral carina distinctly developed, postgenal cushion developed. Toruli at anteriormost position of head, on distinctly protruding antennal shelf. Antennal shelf without any wrinkles. Frons armed with two lateral projections about mid-way between levels of anterior ocellus and toruli. Anteriormost area of frons marked by carinae connecting antennal shelf, lateral projections and anterior ocellus. Additional lateral carina leading from this area diagonally to lateral ocelli and above eyes. Antenna: 12-segmented, elbowed with elongate scape. Scape highly modified, with distinct lateral projection and two sharp corners protruding insertion to pedicel. First flagellomere cylindrical, distinctly longer than subsequent flagellomeres. Clava elongate, distinctly longer than wide. Second and third flagellomere still cylindrical, subsequent flagellomeres spherical. Multiple setal bases present as pores on individual antennomeres. Pronotum adjacent to mesoscutum. Pronotum with long and distinct neck, dorsal pronotal surface short, but visible in dorsal view. Distinct transverse pronotal sulcus present between pronotal neck and dorsal surface. Pronotal neck with irregular striae in posterior half. Posterior margin of pronotum with transverse row of punctures with elongate setae corresponding to distinct internal ridge that articulates with anterior margin of mesoscutum and laterally connects with postspiracular apodemes. Pronotum laterally to neck with distinct patches of short setae. Lateral panel of pronotum large and triangular, adjacent to mesopleuron. Pronotal depression for accommodation of pro-femur absent. Hind corner of pronotum reaching tegula. Mesothoracic spiracles positioned at lateral margin of pronotum, on spike-like protuberances, spiracles posteriorly enclosed by prepectal shelf, dorsally reaching tegula. Mesothoracic spiracles completely enclosed by cuticle. Propleura smooth, posterior margin with narrow rectangular fields. Fields anteriorly and laterally carinate, serving as articulation point for anterior surface of procoxae. Articulation of procoxae further enhanced by two rounded impressions on rectangular prosternum, by distinct procoxal impressions on ventral mesopleuron and by ventrally flattened expansions of pronotum laterally to procoxal foramina. Prosternum transversely divided by complete cross carina. A single median profurcal pit present in extremely narrowed dorsal part of prosternum. Profurca u-shaped, profurcal arms only preserved as thin structures. Propleural arms incompletely preserved.

Mesonotum dorsally flattened. Mesoscutum wider than long, smooth, with very few scattered elongate setae arising from punctures. Notauli present as very broad, curved sulci, which are distinctly dilated posteriorly and deeply pitted anteriorly. Notauli anteriorly nearly reaching anterior mesoscutal margin and posteriorly nearly reaching transscutal articulation. Notauli internally rather weakly marked. Preaxilla smooth, deeply concave with two carinae on surface, arising from anterior and antemedian mesonotal wing processes. Anterior carina articulating with anterior margin of tegula. Tegula huge and rounded, posteriorly reaching posterior wing process. Tegula laterally expanded, with smooth and flattened anterior surface. Tegula with few elongate setae. Transscutal articulation straight and complete. Mesoscutellum with few scattered setae. Scutoscutellar sulcus marked by two large, ovoid pits, which are medially separated by broad straight ridge. Pits internally well defined. Axillae large, triangular, and smooth. Mesoscutellar disc laterally separated from axillula by distinct straight ridges, which are marked by lateral depressions. Hind margin of mesoscutellum distinctly foveolate. Posterior wing process short and broadened, blunt with smooth surface. Mesopleuron smooth and glabrous, mesofemoral depression absent. Mesepimeron with two longi-tudinal ridges connecting anterior and posterior mesopectal margin and separating long rectangular area. Epicnemial pit present with reduced, short pilosity. Sternaulus developed as complete carina connecting epicnemial pit with posterior margin of mesopectus. Mesepisternum anteriorly with distinct procoxal depressions, which are medially separated by short and indistinct carina. Raised acetabular and mesotrochantinal carinae present. On each side posterior margin of acetabular carina connected by longitudinal carina to anterior margin of mesotrochantinal carina. Mesodiscrimen hardly traceable. Two distinct mesofurcal pits developed. Round anterior pit present at acetabular carina, slit-like posterior pit present medially on mesotrochantinal plate just anterior to mesocoxal foramina. Mesocoxal foramina not completely enclosed by cuticle. Meso-discrimenal lamella reaching close to anterior margin of mesopectus. Mesofurca with two solid bases, mesofurcal bridge complete with dorsal orientation.

Metanotum anteriorly overlapped by mesoscutellum. meta-scutellum with two distinctly raised lateral and distinct median carina. Lateral panel of metanotum composed of foveae. Metapleuron subrectangular, mainly smooth, with ventral row of foveae. Metepisternum with distinct depressions for accommodating mesocoxae, and with distinct median carina (corresponding to metadiscrimen). Single metafurcal pit present anteromedially of metacoxal foramina, posterior to raised carina. Metacoxal foramina with lateral projections. Metafurca indistinct, u-shaped, completely fused to highly raised paracoxal ridge. Metadiscrimenal lamella reaching nearly to mid-level of meta-coxal foramina.

Median keel on propodeum formed by v-shaped median carina pointing anteriorly. Anterior margin of propodeum deeply excavate and smooth. Dorsal surface of propodeum with two submedian carinae (instead of single median carina). Plicae developed, dorsal propodeal surface between plicae and sub-median carinae smooth. Posterior margin of propodeum deeply excavate, posterolateral corners strongly projecting and broadly bifurcate. Hind margin of propodeum carinate.

Petiole cylindrical, laterally with long pilosity, 1.49 times as long as wide. Petiole with three dorsal, three lateral and two ventral carinae. Second tergum greatly enlarged, anterior margin with deep and narrow median incision. Second tergum overlapping petiole. Subsequent terga extremely shortened. Second sternum greatly enlarged, covering subsequent three sterna. Second sternum with deep grooves filled with micropilosity.

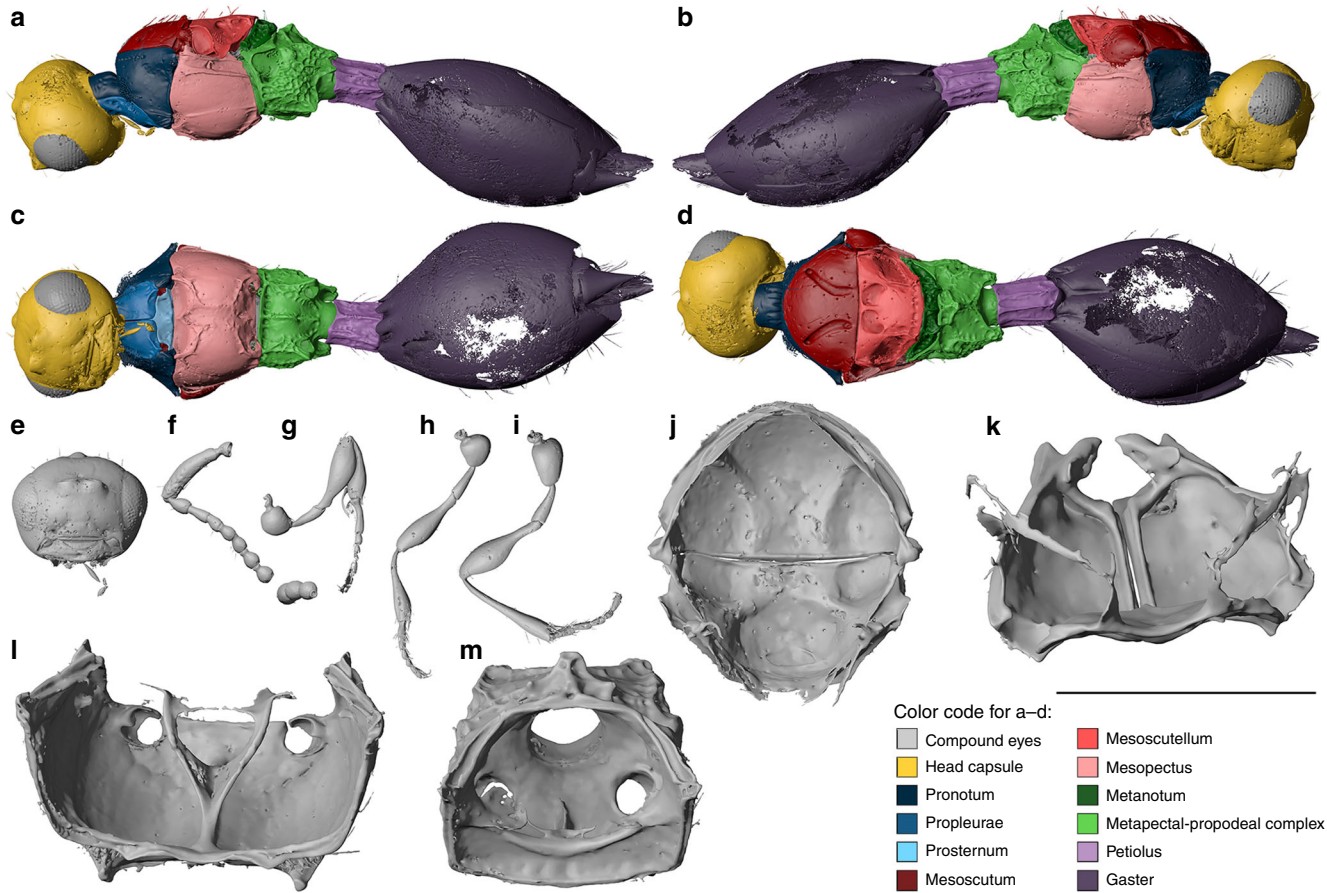

**Fig. 7** Digital reconstruction of *Palaeortona quercyensis* gen. & sp. nov. (Diapriidae: Diapriinae: Psilini). Female holotype NMB F2770. Habitus (**a**–**d**), head (**e**), right antenna (**f**), right foreleg (**g**), right midleg (**h**), and right hind leg (**i**). Internal anatomical structures (**j**–**m**): mesonotum ventral view (**j**), propectus dorsolateral view (**k**), mesopectus anterior view (**l**), and metapectal-propodeal complex anterior view (**m**). Scale bar: **a**–**i** = 1 mm; **j** = 500 μm; **k** = 250 μm; **l**, **m** = 400 μm

Color code for a–d:
- Compound eyes
- Head capsule
- Pronotum
- Propleurae
- Prosternum
- Mesoscutum
- Mesoscutellum
- Mesopectus
- Metanotum
- Metapectal-propodeal complex
- Petiolus
- Gaster

Wings: Not traceable. Legs: All legs with elongate simple trochanters. Protibial spur with distinct cleft. Midleg with two mesotibial spurs. Hind leg with two metatibial spurs.

Male (Fig. 6o–w). Measurements given for paratype: NMB F3154. Very similar to female but differs in following features. Reference length: 652 μm. Head height: 468 μm, head width: 389 μm, head length: 507 μm. IOD:OOD: 0.90. Antenna: 14-segmented. Eyes 224 μm high and 192 μm wide. Foveae on lateral panel of metanotum hardly visible. Petiole 1.83 times as long as wide. T2 hardly overlapping petiole.

Comments: The new species agrees with nearly all generic characters listed by Masner and García[22] for *Coptera* Say, many of which can be regarded as synapomorphies. The only observed morphological difference is the presence of three mandibular teeth in *C. anka* (two in *Coptera*), which we find insufficient to diagnose a new genus.

### *Palaeortona* Krogmann, van de Kamp & Schwermann gen. nov.

**Type species**. *Palaeortona quercyensis* Krogmann, van de Kamp & Schwermann sp. nov.
**Etymology**. The genus name refers to the morphologically similar extant genus *Ortona* Masner and García.
**Diagnosis**. Body without hairy cushions or foamy structures. Labrum exposed, semicircular. Mandible small, not protruding, bidentate. Head not depressed. Oral carina distinct.

### *Palaeortona quercyensis* Krogmann, van de Kamp & Schwermann sp. nov.

**Etymology**. The species epithet refers to the type locality.
**Diagnosis**. Level of toruli lower than midpoint of eye. Pronotal neck distinctly developed. Median ocellus nearly adjacent to lateral ocelli. Petiole cylindrical, 1.3 times as long as wide.
**Referred material**. Holotype ♀: NMB F2770 (Figs. 2 and 7, Supplementary Fig. 4k, Supplementary Fig. 5ad, Supplementary Data 7).
**Locality**. As for *X. resurrecta*.
**Description**. Female (Fig. 7). Reference length: 810 μm. Head sculpture mainly smooth, frons, face, genae and clypeus with scattered punctures. Elongate setae only preserved on frons. Head height: 439 μm, head width: 490 μm, head length: 401 μm. Ocelli large and very close to each other: median ocellus nearly adjacent to lateral ocelli, lateral ocelli medially separated by about their own diameter. IOD:OOD = 0.71. Eyes large, with multiple scattered punctures, eye height: 243 μm, eye width: 192 μm. Malar space very narrow, 58 μm, malar sulcus absent. Occipital carina complete, horseshoe-shaped, anteriorly marked by small punctures. Clypeus narrow, laterally with enlarged anterior tentorial pits, dorsally marked by distinct and curved epistomal sulcus. Mandibles narrow, two-toothed, leaving small, semicircular area from ventral clypeal margin. Area covered by labrum. Toruli dorsally oriented, positioned on moderately protruding antennal shelf, about half-way upon face. Antennal shelf almost

effaced between toruli. Antennal shelf completely smooth without any wrinkles. Antennal shelf not connected to epistomal sulcus by sulci. Supraclypeal area not defined and not expanded. Single median pit present behind toruli. Antenna incompletely preserved. Probably 12-segmented. Left antenna broken off after eighth antennomere, three additional antennomeres preserved. Right antenna broken off after pedicel, five additional antennomeres preserved. Antenna elbowed with elongate scape. Scape surpassing head height. First flagellomere cylindrical distinctly longer than subsequent antennomeres (except clava). Subsequent antennomeres narrowed at base and laterally rounded, each antennomere about as long as wide or slightly longer. Clava about twice as long as previous antennomere. Scape with multiple punctures and few preserved setae. Remaining antennomeres with few scattered punctures.

Pronotum adjacent to mesoscutum. Pronotum with long and distinct neck, dorsal pronotal surface short, not visible in dorsal view. Distinct transverse pronotal sulcus present between pronotal neck and dorsal surface. Pronotal neck with irregular striae in posterior half. Posterior margin of pronotum with transverse row of punctures corresponding to distinct internal ridge that articulates with anterior margin of mesoscutum and laterally connects with postspiracular apodemes. Pronotum laterally to neck with distinct patches of short setae extending to lateral surface of propleura. Lateral panel of pronotum large and triangular, adjacent to mesopleuron. Pronotal depression for accommodation of profemur present. Hind corner of pronotum reaching tegula. Mesothoracic spiracles positioned at lateral margin of pronotum, on spike-like protuberances, spiracles posteriorly enclosed by prepectal shelf, dorsally reaching tegula. Mesothoracic spiracles completely enclosed by cuticle. Propleura smooth, posterior margin with narrow rectangular fields. Fields anteriorly and laterally carinate, serving as articulation point for anterior surface of procoxae. Articulation of procoxae further enhanced by two rounded impressions on rectangular prosternum, by distinct procoxal impressions on ventral mesopleuron and by ventrally flattened expansions of pronotum laterally to procoxal foramina. Prosternum without cross carina. Two submedian pits present in dorsalmost part of prosternum. Profurca u-shaped, profurcal arms only preserved as thin structures. Propleural arms incompletely preserved.

Mesonotum dorsally flattened. Mesoscutum wider than long, smooth, with few scattered elongate setae arising from punctures. Additional setae arranged in lines flanking both sides of each notaulus. Notauli present as very broad, curved sulci, which are distinctly dilated posteriorly. Notauli anteriorly nearly reaching anterior mesoscutal margin and posteriorly nearly reaching transscutal articulation. Notauli internally rather weakly marked. Preaxilla smooth, with two carinae on surface, arising from anterior and antemedian mesonotal wing processes. Anterior carina articulating with anterior margin of tegula. Tegula huge and rounded, posteriorly reaching (and articulating with) axillular carina. Tegula laterally expanded, with smooth and flattened anterior surface. Transscutal articulation straight and complete. Mesoscutellum with few scattered setae. Scutoscutellar sulcus marked by two large kidney-shaped pits, which are medially separated by broad straight ridge. Pits internally rather weakly marked. Axillae narrow and smooth. Mesoscutellar disc laterally separated from axillula by distinct straight ridges. Hind margin of mesoscutellum distinctly foveolate. Posterior wing process broadened and blunt with smooth surface. Mesopleuron smooth and glabrous, mesofemoral depression absent. Mesepimeron with two longitudinal ridges connecting anterior and posterior mesopectal margin and separating long rectangular area. Epicnemial pit present and marked by carina, with dense

pilosity. Sternaulus developed as medially interrupted carinate sulcus connecting epicnemial pit with posterior margin of mesopectus. Mesepisternum anteriorly with distinct procoxal depressions, which are medially separated by short and indistinct carina. Raised acetabular and mesotrochantinal carinae present, both medially interrupted. Each lateral portion of acetabular carina connected by faint longitudinal carina to lateral portion of mesotrochantinal carina. Mesodiscrimen not traceable. Two mesofurcal pits developed. Anterior pit present at interrupted area of acetabular carina, posterior pit present medially on mesotrochantinal plate just anterior to mesocoxal foramina. Mesocoxal foramina not completely enclosed by cuticle. Mesodiscrimenal lamella reaching close to anterior margin of mesopectus. Mesofurca with two solid bases, mesofurcal bridge complete with dorsal orientation.

Metascutellum with two distinctly raised blade-like lateral and less distinct median carina, metascutellum posteriorly expanded. Lateral panel of metanotum composed of anteriorly reduced foveae. Metapleuron anteriorly smooth, posteriorly coarsely reticulate. Metepisternum with distinct depressions for accommodating mesocoxae, and with distinct median carina (corresponding to metadiscrimen). Single metafurcal pit present anteromedially of metacoxal foramina, posterior to raised carina. Metacoxal foramina with lateral projections. Metafurca indistinct, u-shaped, completely fused to highly raised paracoxal ridge. Metadiscrimenal lamella reaching mid-level of metacoxal foramina. Median keel on propodeum formed by v-shaped median carina pointing dorsally. Anterior margin of propodeum deeply excavate and smooth. Plicae developed, dorsal propodeal surface between plicae and median carina smooth. Posterior margin of propodeum deeply excavate, posterolateral corners strongly projecting and bifurcate. Hind margin of propodeum carinate.

Petiole cylindrical, laterally with long pilosity, 1.31 times as long as wide. Petiole with three dorsal, three lateral and one ventral carinae. Second tergum greatly enlarged, anterior margin with deep and broad median incision. Second tergum hardly overlapping petiole. Third and fourth tergum completely covered by second tergum, subsequent terga visible but extremely shortened. Second sternum greatly enlarged, covering at least one of subsequent sterna.

Wings: Unfolded but hardly traceable. Legs: All legs with elongate simple trochanters. Protibial spur with two apices but without distinct cleft. Midleg with two mesotibial spurs. Hind leg with two metatibial spurs.

Male unknown.

Comments: The fossils shares a number of morphological characters with the genus *Ortona* Masner and García[22], which is restricted to the New World. It differs in the following characters: head not depressed, level of toruli lower than midpoint of eye, oral carina distinctly developed, labrum semicircular (not transverse), clava not long ovoid, and pronotal neck distinctly developed. These characters support placement in a new genus.

**Tribal placement**. The current subdivision of Diapriinae into three tribes (Spilomicrini, Psilini, and Diapriini) is supported by morphological data[22]. *Xenomorphia resurrecta* and *X. handschini* can be readily placed in the tribe Spilomicrini based on the presence of a syntergite on the metasoma, the presence of a distinct malar sulcus (faintly indicated only in female of *X. handschini*), the absence of spike-like, protruding mesothoracic spiracles, and the presence of complete notauli. Both species have retained a number of plesiomorphic characters, such as the high number of antennomeres (14) in both sexes, which is otherwise only known from the extant genera *Xenismarus* Oglobin and *Chilomicrus* Masner and García. These genera are

characterized by a restricted, putatively relictual, Valdivian distribution[22]. A macrotergite (second tergum), instead of a syntergite (fused second and third tergum), and the presence of spike-like, protruding mesothoracic spiracles characterize *Coptera anka* and *Palaeortona quercyensis* as members of Psilini, a small tribe that comprises only four extant genera. Only the tribe Diapriini, which is considered the most derived lineage of Diapriinae[22], is not represented in the Quercy fossil parasitoid fauna.

## Discussion

The most common species was *Xenomorphia resurrecta*, of which we found 18 females and 24 males, followed by *X. handschini* with one female, four males and one pupa and *Coptera anka* with three females and one male. *Palaeortona quercyensis* was represented by one female only. Additionally, we identified a single unknown putative second instar wasp larva (Fig. 2, Supplementary Fig. 4l), and a set of last larval instar mandibles (Fig. 2, Supplementary Fig. 4m), presumably left behind by the emerged parasitoid.

The varying quality of preserved wasps (Supplementary Table 1) suggests that not all specimens present at the time of the fossilization are still traceable. Strikingly, 52 out of 55 discovered parasitation events were recognized by the presence of adult wasps, preserved inside the puparia shortly before or after ecdysis. Though soft tissue preservation in the Quercy fossils is known from beetles[23] and amphibians[20,24,25] the preservation in parasitoid wasps seemed to favor sclerotized structures inside the puparia. We hypothesize that the exoskeleton of the adult wasps was more chemically resistant to early postmortal decay than their earlier developmental stages and the host flies. This could explain the higher representation of adult wasps. X-ray diffraction (Methods, Supplementary Fig. 6) confirms that the fossils are composed of phosphate minerals (apatite). It is assumed that the Quercy arthropods fossilized by a rapid fixation by phosphate-rich water followed by encrustation and mineralization[26]. After decay of the cuticle, air-filled cavities were left[23]. In comparison to amber inclusions, which show representational bias towards arboreal taxa[27], the fossil arthropods of the Quercy localities constitute a unique composition of forest floor communities associated with vertebrate carrion[18,23]. Large numbers of specimens from the same species may be found alongside each other[18], offering the potential to obtain not only morphological but also ecological information from this particular ecosystem.

The wasps diagnosed herein all belong to the single family Diapriidae, although various hymenopteran lineages are known to exploit fly hosts in decaying substrates[10]. The extant diapriid fauna comprises more than 2000 described species[28]. Diapriid wasps develop as solitary or gregarious endoparasitoids of fly pupae, while some species develop at the expense of beetles or ants[22]. Only in one specimen did we recover fly legs alongside the parasitoid (Supplementary Fig. 4h). This either means that the respective species (*C. anka*) had the potential to parasitize not only early stages of fly pupae but also almost fully grown adults or that the development of parasitized fly pupae was not immediately paused after oviposition. Unfortunately, the lack of developmental data for extant diapriids hinders a comparison of fossil and extant biology. The observed differences in wing condition, body position and posture indicate a resting period of hatched wasps inside the pupae, putatively resulting from the need for synchronized emergence, a known strategy for insect parasitoids[29]. Based on morphology, four distinct wasp species can be characterized, which were seemingly able to coexist within a forest floor community exploiting fly hosts associated with vertebrate carrion. There is no data on extant diapriid species communities within the same host group in the same habitat but there are many examples from other parasitoid wasp lineages,

including the occurrence of four sympatric species of the genus *Nasonia* parasitizing fly puparia in birds' nests[30,31]. All species described herein are fully winged, indicating the need for dispersal, while several extant diapriid species have their wings reduced in one or both sexes[22]. It seems plausible that there have been differences between the ecological niches of the four species indicated by the striking morphological disparity of species of the tribes Spilomicrini and Psilini. *Coptera anka* and *Palaeortona quercyensis* (both Psilini) are characterized by numerous putatively derived cuticular expansions on the articulation points leading to antennae (modification of scape), wings (enlargement of tegula) and petiole (lateral expansion of propodeum) serving as protections for the concerned articulation points. With these characters they would be better equipped than the two *Xenomorphia* species (Spilomicrini) for a ground-dwelling lifestyle as an adaptation to more concealed hosts. The head expansions of *C. anka* further facilitate such a forward-directed movement through leaf litter and other ground associated material.

The evolutionary history of Diapriinae is largely unresolved due to the scarcity of well-preserved fossils[32–34], and the absence of a robust phylogenetic hypothesis for the whole family. From a phylogenetic perspective it is relevant to note that the morphological body plan represented by the fossil *C. anka* remained largely unaltered over a period of about 30 million years, while the other three fossil species represent morphological concepts that have either been significantly modified among extant descendants, or that represent evolutionary dead ends. This implies that the extant diapriine wasp fauna comprises species that exhibit varying degrees of ecological niche conservation, i.e., the retention of ecological characters over evolutionary time scales[35,36]. The ecological and morphological data preserved in the fossil host–parasitoid complex described herein will provide the basis for future comparative studies. It also highlights the need for closing the existing knowledge gap of the morphological and biological diversity of extant parasitoid wasps.

## Methods

**Samples**. The fossils originate from the collections of the Natural History Museum of Basel (NMB) and the Swedish Museum of Natural History (NRM), where all holo- and paratypes of the current study have been deposited. They were collected in the phosphorite mines of the Paleogene fissure fillings of the Quercy region in South-Central France, but the exact locality, collection date and the original collector are unknown. Given the information provided by Handschin[18], it is likely that the samples housed in Basel were acquired mainly by the fossil collector Rossignol around 1900, who sold them to the Natural History Museum of Basel. This collection was extended by specimens picked by Stehlin and Helbing. The *Eophora*-specimens were discovered near Bach, which is also true for *Spiniphora*-specimens, whereas *Megaselia*-specimens were also collected near Caylux[18]. The Quercy collection housed in the Swedish Museum of Natural History is mainly based on a donation from the Zoological Institute (the former Zootomical Institution) of the Stockholm University, made in the 1960s to the museum. This donation consisted of two larger collections, one bought from Rossignol, with collection dates ranging from 1890 to 1906. The other collection was made before 1883 by Kowalewski. Additional material was bought from the fossil collectors Dagincourt in 1886, Stürtz in 1894, Krantz (former collection Filhol) in 1904 and 1906, and Stuer in 1906. Combined, the collections contain a total of 1510 individual fly pupae, which can be assigned to three different morphospecies. Those were defined by Handschin[18] as belonging to the genera *Eophora* Handschin (unavailable name[21]), *Megaselia* Rondani, and *Spiniphora* Malloch. The Basel collection contains 1188 *Eophora* (collection numbers NMB F2441-F3628), 37 *Megaselia* (NMB F3629-F3665), and 14 *Spiniphora* (NMB F3666-F3679); the Stockholm collection is comprised of 252 *Eophora* (NRM-PZ Ar65716-Ar65967), 18 *Megaselia* (NRM-PZ Ar65768-Ar65985), and one *Spiniphora* (NRM-PZ Ar65786).

**Photography**. Z-stack photographs of the puparia were acquired with a Zeiss Axio Zoom.V16 microscope (Carl Zeiss AG, Oberkochen, Germany) equipped with a PlanApo Z 1.0 × /0.25 FWD 60 Objective, a CL 6000 LED Illumination and an AxioCam HRc Camera. Images were processed with the software Zen 2 and Photoshop (Adobe Systems Incorporated, San José, USA) by cropping, contrast enhancement and the removal of the background.

**High-throughput synchrotron X-ray microtomography**. Tomographic scans were performed at the UFO imaging station of the KIT light source. A parallel polychromatic X-ray beam produced by a 1.5 T bending magnet was spectrally filtered by 0.2 mm Al to remove low energy components from the beam. The resulting spectrum had a peak at about 15 keV, and a full-width at half maximum bandwidth of about 10 keV. The beamline was equipped with an automated sample change robot (Advanced Design Consulting USA, Inc.) and a fast indirect detector system consisting of a 12 μm LSO:Tb scintillator[37], diffraction limited optical microscope (Optique Peter) and 12 bit pco.dimax high speed camera with 2016 × 2016 pixels resolution[38]. We used the control system concert[39] for automated data acquisition. Fast screening of all samples was performed with an optical magnification of 5×, resulting in an effective pixel size of 2.44 μm. For screening, we took 1500 or 2000 equiangularly spaced radiographic projections in a range of 180° at 70 fps. Screening data were evaluated online during radiographic data acquisition and after reconstruction of the respective tomographic volumes. Samples containing presumptive parasitoids or hosts were selected for additional high-resolution scans. These were done by taking 3000 projections at 70 fps and an optical magnification of 10× (1.22 μm effective pixel size). Due to a smaller field of view at high magnification, anterior and posterior portions of each puparium were scanned separately. Both tomograms were subsequently registered and stitched in Amira 5.6. Tomographic reconstruction was performed with a GPU-accelerated filtered back projection algorithm implemented in the UFO software framework[40]. With the exception of semiautomated segmentation (see below), postprocessing of tomographic data was largely performed using the ASTOR virtual analysis infrastructure at KIT[41].

**Volume rendering**. The stitched high-resolution tomograms were aligned and cropped using Amira 5.6. For visualization of the inclusions (Fig. 1d–f, m; Supplementary Figs. 2–4), they were subsequently inverted and the puparia were isolated from the background using the software's segmentation editor. The labelfields were saved as TIF stacks and served for masking the background of the inverted datasets in Fiji[42]. Volume rendering of all postprocessed datasets was performed in Drishti 2.5.1[43].

**Semiautomated segmentation and creation of polygon meshes**. High-resolution tomograms were imported into Amira 5.6. Important structures were presegmented in the software's segmentation editor by manually labeling every tenth slice. Distinct morphological structures were assigned to different "materials". The labels served as input for automated segmentation, which was done using the web application Biomedisa (https://biomedisa.de), developed by one of the authors (P.D.L.). Its segmentation process is based on a highly scalable diffusion method, which is free of hyperparameters and thus eliminates an elaborate and tedious configuration[44]. Segmentation results were imported in Amira 5.6 and all individual parts were converted into polygon meshes by employing the "SurfaceGen" tool. The meshes were exported as OBJ files and reassembled in CINEMA 4D R18, which was employed for polygon reduction and smoothing. The data were saved in the DAE format, imported into Deep Exploration 6 and converted into U3D files. The latter were embedded into PDF documents using Adobe Acrobat 9 Pro Extended[45].

**Illustration**. The true-to-life impression (Fig. 3, Supplementary Movie 2) was created by rearranging the original mesh of the *Xenomorphia resurrecta* holotype (NMB F2875) and placing it on top of the mesh of a puparium (NRM-PZ Ar65767). Rearrangement, animation and rendering were done in CINEMA 4D 18. Coloration was finalized using Adobe Photoshop.

**Micro X-ray diffraction**. Five randomly selected specimens (NMB F2459, NMB F2460, NMB F2531, NMB F2557, and NMB F2915) were characterized by micro X-ray diffraction (μ-XRD) at the SUL-X beamline of the KIT light source. Puparia were placed in glass capillaries with a diameter of 1.5 mm. Diffraction data measurements were recorded with a charge-coupled device (CCD) detector (Photonic Science XDI VHR-2 150) in symmetric transmission at 17 keV in focused beam (ca. 100 μm vertical and 250 μm horizontal at sample position). A sample-detector distance of about 110 mm and a detector area of 80 × 120 mm resulted in a maximum diffraction angle of ca. 35° (lattice spacing d ca. 1.2 Å), sufficient for mineral identification. CCD frames were analyzed using Fit2D[46,47]. Instrumental parameters were determined with a LaB6 NIST Standard (660b) and subsequently employed for integration of the sample frames into 1D diffractograms (Supplementary Fig. 6). For mineral identification the ICDD database (www.icdd.com; PDF-2) has been used.

**Nomenclatural acts**. This published work and the nomenclatural acts it contains have been registered in ZooBank, the proposed online registration system for the International Code of Zoological Nomenclature. The ZooBank LSIDs (Life Science Identifiers) can be resolved and the associated information viewed through any standard web browser by appending the LSID to the prefix "http://zoobank.org/". The LSIDs for this publication are: urn:lsid:zoobank.org:act:6B023E01-517B-4017-83E3-941477C87148, urn:lsid:zoobank.org:act:E2A9A7A4-63C6-45CB-B921-80C324309461, urn:lsid:zoobank.org:act:D773F715-948E-4D22-9A29-3FD5F0FB767E, urn:lsid:zoobank.org:act:01D04027-1DDB-453E-A286-

8D4B5AD58CF6, urn:lsid:zoobank.org:act:161C703D-B85F-489B-AC00-3161B5F01567, and urn:lsid:zoobank.org:act:DBBE09E5-111A-4225-9488-7A7C955BE829.

**Data availability**. CT-raw data were generated at the imaging cluster of the KIT light source. Derived data supporting the findings of this study are available from the corresponding authors upon reasonable request. High-resolution datasets showing parasitation events are deposited at http://www.fossils.kit.edu.

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

## Acknowledgements

We thank Erik Åhlander for information about the accession of the fossils, Georg Oleschinski for taking the photographs and Stephen Doyle for improving the language of the manuscript. Heiko Schmied was involved in the early stage of the project. István Mikó, Lubomir Masner, and Marina Moser are thanked for helpful comments and fruitful discussions. We acknowledge the KIT light source for provision of instruments at their beamlines and we would like to thank the Institute for Beam Physics and Technology (IBPT) for the operation of the storage ring, the Karlsruhe Research Accelerator (KARA). Analytical tools used in this study were provided by the projects ASTOR and NOVA (Michael Heethoff, TU Darmstadt; Vincent Heuveline, Heidelberg University; Jürgen Becker, KIT), funded by the German Federal Ministry of Education and Research (BMBF; 05K2013, 05K2016). We especially thank the following colleagues: Felix Beckmann, Jörg Hammel, Wolfgang Mexner, and Sebastian Schmelzle. Research at KIT was further supported by the project UFO 2 (BMBF; 05K2012). We acknowledge support by Deutsche Forschungsgemeinschaft and Open Access Publishing Fund of Karlsruhe Institute of Technology.

## Author contributions

T.v.d.K. and A.H.S. designed the study. W.E., T.M. B.M., and J.R. located and identified specimens. T.v.d.K., A.H.S., T.d.S.R., T.E., J.O., T.F., L.K., P.D.L., and T.B. performed μCT measurements. T.F. and M.V. developed data acquisition and reconstruction protocol. T.F. and T.d.S.R. reconstructed μCT data. T.v.d.K., A.H.S., T.E., J.O., and L.K. analyzed μCT data. P.D.L. and V.H. developed semiautomated segmentation tools. T.v.d.K. created volume renderings, surface models and illustrations. J.G. conducted X-ray diffraction experiments. N.T.J. and A.K. created database and online portal. L.K. performed species descriptions and systematic placement. T.v.d.K., A.H.S. and L.K. drafted the manuscript. All authors contributed to the writing and discussion.

## Additional information

**Competing interests:** The authors declare no competing interests.

