## [Peer Review File · Nature Communications]

REVIEWERS' COMMENTS:

Reviewer #1 (Remarks to the Author):

The manuscript offers documentation of a host-parasitoid relationship in the fossil record. Such relationships have occasionally been proffered in the past, but in my experience these were based solely on the physical proximity of specimens of the purported partners. The observation of pupae and newly emerged adults within the fly puparia is likely as near as we will come to certainty of their ecological interaction. The methods used to visualize these minute wasps within the fly puparia have resulted in images that are astounding for their detail. The fact that these results are based upon dozens of specimens found by searching over 1500 puparia is impressive and very welcome for a paleontological study. The taxonomic placement and treatment of the wasps appears to be sound and well documented.

Overall, my assessment of this manuscript is very positive. I would, however, offer the following criticisms. First, and most important, the authors have failed to explicitly designate a type species for the new genus *Xenomorphia*. Such an explicit designation is a requirement for the name to be available under Article 13.3 of the International Code of Zoological Nomenclature. Additionally, I would recommend that the single included species in *Palaeortona* also be explicitly designated as the type of the new genus.

In describing the biology of Diapriidae, the authors state that their hosts are member of the order Diptera with exceptional species developing at the expense of beetles. I must point out that ants are also known hosts. Although the number of documented host records for those genera is small, some of these genera (such as *Acanthopria*) are very speciose. Thus, probably more than the "exceptional" species is a parasitoid of something other than a fly. As a further note, the authors state that the previous attribution of a role as parasitoid in fossil diapriids was only "assumed." Perhaps I am simply misinterpreting their use of this word, but I believe that such an attribution was a sound inference made on the basis of phylogenetic relationship and parsimony. To me, the word "assumed" can carry the implication that the assumption is not supported in any way, and this is not correct.

The observation of numerous specimens that have eclosed from the pupal stage, but have not yet emerged from their host is used to support the idea of synchronous emergence in the wasps. That may be true, but I believe that there are other interpretations consistent with the same data. The most obvious one is that the adult wasps remain within the host puparium for a period of time as their cuticle hardens and darkens after eclosion. Further, the window of time during which the host is suitable for parasitization may be quite small. As a result, wasps parasitizing flies in the same substrate will experience largely similar environmental conditions, particularly temperature, and any synchronization may result from that rather than the implied role of adult wasps "waiting" for the proper time to emerge.

Finally, the authors cite the genus *Nasonia* as a striking "occurrence of several sympatric species" in the same microhabitat. There are only four known species of *Nasonia*, and so I think the use of the word several may imply a significantly larger number of species to a reader not familiar with that fact. I would also like to note that the two reference cited in support of this observation about *Nasonia* are not independent. The second reference (#27) simply cites the findings of the first.

Reviewer #2 (Remarks to the Author):

This manuscript provides an exciting new application for SR X-ray micro-CT scanning to examine a historical collection of fossil fly puparia and shed new light on their diapriid wasp parasitoids. The work provides thorough descriptions of the parasitoid taxa, establishing a paleoecological relationship that is seldom observed in the fossil record, and bringing an unprecedented level of

detail to the study of this association. The major claims of the paper are in general well-supported, and the systematic work is exceptional -- particularly considering the type of preservation involved. This study highlights the potential for SR X-ray micro-CT, and will likely inspire researchers to revisit other fossil collections with advanced imaging techniques. I recommend the manuscript for acceptance after minor revisions.

Aside from the minor annotations in the attached PDF files (most of which are minor suggestions for wording or grammar), there was only one taphonomic issue that is worth considering. The inference that the diapiids are predominantly adults resting prior to a synchronous flight from the puparia is well-supported by the observed morphology and body positions, but the relative proportion of adult specimens hinges on the assumption that both the adults and immatures have an equal preservation potential. It seems very likely that the scan data are biased toward recovering adult parasitoids, because these are the specimens that are most likely to have thick cuticle capable of leaving a void behind and creating strong contrast in the scans. This issue can probably be addressed with a brief comment about the potential for bias, which may impact both the overall incidence of parasitism, and the ratio between adults and immatures in the sample set.

Reviewer #3 (Remarks to the Author):

The work under review is impressive and important. The authors examined with X-rays a voluminous and rare material of numerous (1510) ancient (Eocene) fly puparia from Quercy, France, phosphatized in situ among remains of vertebrate corpses the fly larvae fed on. X-ray examination revealed 55 puparia parasitized with four species of diapiid wasps of terrific preservation state permitting reliable identification of genus and description of species. The taxonomic results reveal presence of four new species in three genera: one new genus of basal position within the family and similar to two rare living genera of Valdivian distribution (two new species), another new genus with one new species, also of rather basal position and related to a living Nearctic genus, and the third one, an advanced living genus of Nearctic distribution (one species). The results shed unexpected and important light to the Paleogene history of Diapriidae including their already high diversity, changed distribution and apparently stable biology. Particularly striking are illustrations that use all achievements of the modern techniques. The results are original, convincing, novel and will raise considerable general interest as an example of great possibilities that the modern research techniques open in the area still often dominated by traditional approaches. The work unquestionably deserves quick publication.

The comments are not numerous, as follows.

P. 3 and in general: There are many students exist (me included) that feel indication of a position of deposits on the stratigraphic scale (this case the later Middle to Late Eocene) as more meaningful and comfortable than just to refer to the absolute age. So I would suggest combining both form of indication at least at the first mention of the fossil site

P. 3 and others: An interesting characteristics of Quercy environment and taphonomy is provided by V.V. Zherikhin (in Rasnitsyn, Quicke. 2002. History of Insects, Dordrecht, Kluwer Acad. Publ., p. 384): I would suggest considering it as well

P. 3, ln. 68: Neal L. Evenhuis (<http://hbs.bishopmuseum.org/fossilcat/fossporidae.html>) writes: EOPHORA Handschin
EOPHORA Handschin, 1944: 16. Unavailable name; genus-group name proposed after 1930 without named included species.

This opinion should be considered in the paper in a way

P. 4, ln.68: Supplementary discussion (lns 189-194) provides only incomplete comparison of the new genus to the two living genera which are similar in having antenna 14-segmented: no comparison is given to *Chilomicrus* that also has antenna 14-segmented

Alex Rasnitsyn

Response to reviewers' comments

Reviewer #1 (Remarks to the Author):

The manuscript offers documentation of a host-parasitoid relationship in the fossil record. Such relationships have occasionally been proffered in the past, but in my experience these were based solely on the physical proximity of specimens of the purported partners. The observation of pupae and newly emerged adults within the fly puparia is likely as near as we will come to certainty of their ecological interaction. The methods used to visualize these minute wasps within the fly puparia have resulted in images that are astounding for their detail. The fact that these results are based upon dozens of specimens found by searching over 1500 puparia is impressive and very welcome for a paleontological study. The taxonomic placement and treatment of the wasps appears to be sound and well documented.

Overall, my assessment of this manuscript is very positive. I would, however, offer the following criticisms.

1. First, and most important, the authors have failed to explicitly designate a type species for the new genus *Xenomorphia*. Such an explicit designation is a requirement for the name to be available under Article 13.3 of the International Code of Zoological Nomenclature. Additionally, I would recommend that the single included species in *Palaeortona* also be explicitly designated as the type of the new genus.

Thank you for this important note. Type species are now designated for *Xenomorphia* and *Palaeortona*.

2. In describing the biology of Diapriidae, the authors state that their hosts are member of the order Diptera with exceptional species developing at the expense of beetles. I must point out that ants are also known hosts. Although the number of documented host records for those genera is small, some of these genera (such as *Acanthopria*) are very speciose. Thus, probably more than the "exceptional" species is a parasitoid of something other than a fly.

Agreed. We have replaced the term "exceptional species" by "some species" and now reference the occurrence of ants as hosts.

3. As a further note, the authors state that the previous attribution of a role as parasitoid in fossil diapriids was only "assumed." Perhaps I am simply misinterpreting their use of this word, but I believe that such an attribution was a sound inference made on the basis of phylogenetic relationship and parsimony. To me, the word "assumed" can carry the implication that the assumption is not supported in any way, and this is not correct.

Agreed. The negative word "assumption" has been changed by the more neutral word "inference".

4. The observation of numerous specimens that have eclosed from the pupal stage, but have not yet emerged from their host is used to support the idea of synchronous emergence in the wasps. That may be true, but I believe that there are other interpretations consistent with the same data. The most obvious one is that the adult wasps remain within the host puparium for a period of time as their cuticle hardens and darkens after eclosion. Further, the window of time during which the host is suitable for parasitization may be quite small. As a result, wasps parasitizing flies in the same substrate will experience largely similar environmental conditions, particularly temperature, and any synchronization may result

from that rather than the implied role of adult wasps “waiting” for the proper time to emerge.

The environmental conditions that the reviewer postulates can indeed be the reason for synchronized emergence. However, we do not see how this is contradictive to what we wrote in the paper. We do not know what triggered the putative synchronized emergence, but the observed body posture of parasitoids indicates a resting period which can be the result of climatic conditions or something else. As we do not know what exactly triggered this resting period, we find it more cautious to simply state that the fossil parasitoids exhibited this resting period, a pattern also found in extant parasitoids as indicated by the reference we provide.

5. Finally, the authors cite the genus *Nasonia* as a striking “occurrence of several sympatric species” in the same microhabitat. There are only four known species of *Nasonia*, and so I think the use of the word several may imply a significantly larger number of species to a reader not familiar with that fact. I would also like to note that the two reference cited in support of this observation about *Nasonia* are not independent. The second reference (#27) simply cites the findings of the first.

Thanks for pointing this out. We have replaced the word “several” by “four” to give the exact number of known sympatric *Nasonia* species. Also the reference that we erroneously listed under #27 is indeed not adding information to our discussion. We have added the correct reference now, which was also written by Raychoudhury, R. *et al.* in the same volume of *Heredity*, but the correct title is “Behavioural and genetic characteristics of a new species of *Nasonia*”.

Reviewer #2 (Remarks to the Author):

This manuscript provides an exciting new application for SR X-ray micro-CT scanning to examine a historical collection of fossil fly puparia and shed new light on their diapriid wasp parasitoids. The work provides thorough descriptions of the parasitoid taxa, establishing a paleoecological relationship that is seldom observed in the fossil record, and bringing an unprecedented level of detail to the study of this association. The major claims of the paper are in general well-supported, and the systematic work is exceptional -- particularly considering the type of preservation involved. This study highlights the potential for SR X-ray micro-CT, and will likely inspire researchers to revisit other fossil collections with advanced imaging techniques. I recommend the manuscript for acceptance after minor revisions.

1. Aside from the minor annotations in the attached PDF files (most of which are minor suggestions for wording or grammar), there was only one taphonomic issue that is worth considering. The inference that the diapiids are predominantly adults resting prior to a synchronous flight from the puparia is well-supported by the observed morphology and body positions, but the relative proportion of adult specimens hinges on the assumption that both the adults and immatures have an equal preservation potential. It seems very likely that the scan data are biased toward recovering adult parasitoids, because these are the specimens that are most likely to have thick cuticle capable of leaving a void behind and creating strong contrast in the scans. This issue can probably be addressed with a brief comment about the potential for bias, which may impact both the overall incidence of parasitism, and the ratio between adults and immatures in the sample set.

We followed the helpful suggestions and corrections given in the attached PDF. We fully agree that adult and immature wasps do not necessarily have an equal preservation potential. This is mentioned later in the paper where we discuss the potential bias towards more heavily sclerotized adult wasps

in the fossil record of Quercy. By adding another sentence we further highlight the fossil preservation bias towards adults.

Reviewer #3 (Remarks to the Author):

The work under review is impressive and important. The authors examined with X-rays a voluminous and rare material of numerous (1510) ancient (Eocene) fly puparia from Quercy, France, phosphatized in situ among remains of vertebrate corpses the fly larvae fed on. X-ray examination revealed 55 puparia parasitized with four species of diapiiid wasps of terrific preservation state permitting reliable identification of genus and description of species. The taxonomic results reveal presence of four new species in three genera: one new genus of basal position within the family and similar to two rare living genera of Valdivian distribution (two new species), another new genus with one new species, also of rather basal position and related to a living Nearctic genus, and the third one, an advanced living genus of Nearctic distribution (one species). The results shed unexpected and important light to the Paleogene history of Diapriidae including their already high diversity, changed distribution and apparently stable biology. Particularly striking are illustrations that use all achievements of the modern techniques. The results are original, convincing, novel and will raise considerable general interest as an example of great possibilities that the modern research techniques open in the area still often dominated by traditional approaches. The work unquestionably deserves quick publication.

The comments are not numerous, as follows.

1. P. 3 and in general: There are many students exist (me included) that feel indication of a position of deposits on the stratigraphic scale (this case the later Middle to Late Eocene) as more meaningful and comfortable than just to refer to the absolute age. So I would suggest combining both form of indication at least at the first mention of the fossil site

We agree and have added the stratigraphic scale.

2. P. 3 and others: An interesting characteristics of Quercy environment and taphonomy is provided by V.V. Zherikhin (in Rasnitsyn, Quicke. 2002. History of Insects, Dordrecht, Kluwer Acad. Publ., p. 384): I would suggest considering it as well

Thanks for pointing this out. We have added this interesting reference.

3. P. 3, ln. 68: Neal L. Evenhuis (<http://hbs.bishopmuseum.org/fossilcat/fossphoridae.html>) writes:
Genus EOPHORA Handschin
EOPHORA Handschin, 1944: 16. Unavailable name; genus-group name proposed after 1930 without named included species.
This opinion should be considered in the paper in a way

We have referenced the catalogue and now explicitly mention that *Eophora* is an unavailable name.

4. P. 4, ln.68: Supplementary discussion (lns 189-194) provides only incomplete comparison of the new genus to the two living genera which are similar in having antenna 14-segmented: no comparison is given to *Chilomicrus* that also has antenna 14-segmented

We appreciate this comment and have now added the morphological characters that clearly separate *Xenomorphia* from *Chilomicrus*.